# Domestic Work, Self-Reported Diagnosed Depression and Related Costs among Women and Men—Results from a Population-Based Study in Sweden

**DOI:** 10.3390/ijerph18189778

**Published:** 2021-09-16

**Authors:** Anu Molarius, Alexandra Metsini

**Affiliations:** 1Centre for Clinical Research, Region Värmland, 651 85 Karlstad, Sweden; 2Department of Public Health Sciences, Karlstad University, 651 88 Karlstad, Sweden; 3Department of Sustainable Development, Region Värmland, 651 82 Karlstad, Sweden; alexandra.metsini@regionvarmland.se; 4Institute of Medical Sciences, Örebro University, 701 82 Örebro, Sweden

**Keywords:** domestic work, gender, mental health, population studies, Sweden

## Abstract

Background: In contrast to paid work, few studies have investigated the association between unpaid domestic work and mental health. The aim of this study was to investigate the association between domestic work and self-reported diagnosed depression and to estimate related costs in a general population. Method: The study is based on women (N = 7981) and men (N = 6203) aged 30–69 years who responded to a survey questionnaire in Mid-Sweden in 2017 (overall response rate 43%). Multivariate logistic regression models, adjusting for age group, educational level, family status, employment status, economic difficulties, and social support, were used to study the association between domestic work and depression. The estimation of direct and indirect costs was based on the calculation of population attributable risks, the literature, and administrative data. Results: In total, 25% of the women and 14% of the men spent more than 20 h a week on domestic work, and 57% of the women and 39% of the men experienced domestic work sometimes or more often as burdensome. A strong independent association between experiencing domestic work as burdensome and depression was observed both in women and men. The total cost of depression possibly related to burdensome domestic work was estimated up to EUR 135.1 million (min EUR 20.7 million–max EUR 21.4 billion) of the total EUR 286.4 million per year in Mid-Sweden. Conclusions: The association between experiencing domestic work as burdensome and depression was strong among both women and men and was not restricted to employed persons or to parents with children. Even though the cross-sectional design does not allow one to assess the direction of the association between domestic work and depression, and longitudinal studies are needed, the results imply that strain in domestic work should be taken into account when considering factors that contribute to the prevalence of depression in the general population and its high societal costs.

## 1. Introduction

Numerous studies have investigated the effects of employment and working conditions on mental health, and these are well documented in the occupational health literature. The effects of unpaid domestic work have been much less studied [1] even though domestic work has a large impact on the everyday life of millions of people. According to the WHO report on social determinants of health from 2008 [2], childcare responsibilities are the most important barrier to participation in the waged labour market for many women throughout the world. Even in countries where women participate in the waged labour market, they continue to be responsible for a major part of the duties for childcare and unpaid work in the household [3].

In Sweden, men spend more time on domestic work and childrearing than men in many other countries do [4]. Still, the amount of time spent on domestic work is considerably higher among women than among men [5]. The most time on domestic work, about 40 h/week, is spent by women with small children. In addition, single mothers are a particularly disadvantaged group, since they face a heavier burden of domestic work and poorer health than non-single mothers [6,7]. Even single fathers have been found to have poorer self-rated health and mental health than non-single fathers [8].

In the past, different theories have been put forward to explain the effect of domestic work on women’s health [9]. According to *t**he double burden theory*, combining paid work with unpaid domestic work is burdensome for women and leads to poorer health. *The expansion theory*, on the other hand, claims that the double roles of women in the family and in the labour market provide a broader social network, financial security and higher self-esteem and are therefore beneficial for women’s health. A limitation of both the expansion and the double burden theories is that they only refer to women.

Other theories that link domestic work to health, and are not limited to women, include the notion that domestic work consists of low-prestige activities that are also physically demanding, routine and isolating which leads to lower levels of mental well-being [1]. Another explanation is the concept of role strain or overload which implies that human energy is limited and that the more demands or roles, the greater the negative effects on health—a notion that is close to the double burden theory [1,9,10]. A third explanation focusses on the relative proportion of domestic work performed in relation to one’s partner and the perceptions on equity in a relationship [1,11,12]. This explanation does, however, not apply to persons living alone or to single parents. In addition, there is a large body of more recent studies on work-family conflicts indicating that conflicts between the demands in the family and work spheres have negative effects on health among employed women [13,14,15] and men [14], but few studies have concerned domestic work in and of itself [1,15,16,17].

The previous literature on the association between domestic work and mental health has produced some conflicting findings, with several studies finding an association between domestic work and mental health problems in both men and women [1,10,13,14,18], whereas in other studies no or weaker associations were found in men [19,20]. In many previous studies of the association between domestic work and health, the study population has only included women [12,15,16,21,22].

Depression is the leading cause of disability worldwide and a major contributor to the global disease burden according to the WHO [23]. The economic burden of depression is high, both to the individual as well as to the wider society. In 2005, indirect costs were estimated at EUR 3 billion (86% of total costs) and direct costs at EUR 500 million (14%) in Sweden [24]. In the same study, the cost of drugs was estimated at EUR 100 million (3% of total costs). The total cost per patient in primary care has been estimated at EUR 5500 (95%CI EUR 5000–6100) over six months in 2005 prices [24]. Direct costs were estimated at EUR 1900 (EUR 1700–2200), 35% of total costs, and indirect costs at EUR 3600 (EUR 3100–4100), 65% of total costs. The cost for antidepressants accounts for 4% of the total costs [25].

A study in 2007 estimated the societal per-patient cost of depression in specialised psychiatric care in Sweden to be around EUR 17,279 on average per year [26]. Production loss costs (sick leave, early retirement) are the largest part of costs (88%), followed by outpatient costs (6%) and drug costs (3%) [24,26]. To our knowledge, no previous studies have tried to estimate the direct and indirect economic burden of depression related to domestic work.

Since there are few studies concerning the general population where both men and women with different family statuses and different employment statuses are included, the aim of this study was to investigate the association between unpaid domestic work and self-reported diagnosed depression among women and men aged 30–69 years in the general population in Mid-Sweden. Due to the high societal costs of depression and lack of studies estimating costs related to domestic work, we also wanted to provide an example on how to estimate the relevant economic burden to the health care system and to society.

## 2. Materials and Methods

The current study is based on a postal survey questionnaire sent to a random population sample aged 18 years and older during March–May 2017. The aim of the survey was to monitor health and its determinants in the general population. The data collection was completed after two postal reminders. Questions on domestic work were posed only in the age group 30–69 years. A total of 14,184 subjects in this age group answered the questionnaire. The overall response rate was 43%. The area investigated covers 55 municipalities in five counties with more than one million inhabitants in Mid-Sweden.

The individuals in the sample were informed that responded questionnaires would be linked to the Swedish official registries through the personal identification numbers, to achieve information on gender, age, educational level and country of birth. The respondents thus accepted the linking of registry data by informed consent. Statistics Sweden carried out the sampling, collected the data and performed the linkage with register data. After the application of registry data, the material was unidentified before it was sent to the counties/regions for further processing. All procedures involving human subjects were approved by the Regional Ethics Board in Uppsala (EPN 2015/417). The data material is protected according to the Public and Privacy Act (2009:400, Chapter 24, Section 8).

### 2.1. Outcome and Major Risk Factors

*Self-reported diagnosed depression* was assessed with the following question: “Do you have any of the following diagnosed illnesses?” where depression was one of the illnesses listed (with answer options yes/no).

There were two questions about *domestic work*. The first one was: “How many hours a week on average do you spend working at home that is not paid work? E.g. taking care of children, nursing relatives, buying the groceries, cooking, paying the bills, washing the laundry, cleaning, taking care of a car, a house or a garden”. The second question asked how often the respondent experienced domestic work as burdensome (never, seldom, sometimes, all or most of the time). The alternatives were dichotomised into “never/seldom” and “sometimes or more often”.

### 2.2. Covariates

*Educational level* was obtained from a national education register and categorised into three levels: low (elementary school), medium (upper secondary school) and high (at least 3 years of university or corresponding education). *Family status* was obtained from a survey question and categorised into living alone, living with partner, living with partner and children, single parent and other. *Employment status* was derived from a survey question about whether the respondent was employed (including self-employed), a student, unemployed or other.

*Economic difficulties* were estimated with the question “During the last 12 months, have you ever had difficulty in managing the regular expenses for food, rent, bills etc.?” (“no”, “yes, once”, “yes, more than once”). The alternatives were dichotomised into “no” and “yes”.

*Social support* was assessed with the question “Do you have anyone you can share your innermost feelings with and confide in?” (yes/no).

### 2.3. Statistical Analyses

The analyses were performed separately for men and women. The associations between domestic work and self-reported diagnosed depression were analysed using bivariate and multivariate logistic regressions. In the first step, bivariate associations between time spent on domestic work and depression as well as burdensome domestic work and depression were calculated. In the second step, time spent on domestic work and burdensome domestic work were included in the same model, and age group, educational level, family status, employment status, economic difficulties and social support were treated as potential confounders and adjusted for. The results are reported as odds ratios (OR) and 95 percent confidence intervals (95% CI) for depression. For further investigation, a similar analysis was carried out among non-employed respondents only and another analysis excluding parents with children from the total study population. These analyses were adjusted for the potential confounders and for gender.

To calculate the proportion of total prevalence of self-reported diagnosed depression that could be attributed to experiencing domestic work at least sometimes as burdensome, population attributable risks (PAR) were calculated using the following formula:PAR = p(OR − 1)/(p(OR − 1) + 1)
where p is the proportion of exposed (prevalence of burdensome domestic work) in the population, and OR is the corresponding odds ratio from the logistic regression analysis.

### 2.4. Estimation of Costs

The cost of depression per year and the cost possibly linked to domestic work were estimated per sex per individual and for the entire population in Mid-Sweden. First, the cost of depression was estimated by calculating the direct and indirect costs. The direct costs were calculated as the sum of health care costs and pharmaceutical spending. Health care costs per patient were obtained from the Cost-Per-Patient database (Region Värmland) for 2018. The cost of pharmaceuticals (the most often used for the treatment of depression: selective serotonin reuptake inhibitors (SSRIs)) was estimated based on pharmacies’ selling price (AUP) per tablet [27]. Population data for the age group of 30–69 years in 2018 in Mid-Sweden were obtained from Statistics Sweden [28]. Indirect costs represent the cost of lost production and was estimated based on the published literature which estimated that indirect costs are around 65% of the total costs [24]. The cost of depression possibly linked to domestic work was finally estimated by multiplying the cost of depression (per sex, per patient and total population) with the population attributable risks calculated in the previous analysis. All costs are presented as average and minimum-maximum values in EUR. The exchange rate (Nov 2020) used was SEK 1.00 = EUR 0.098521.

Since the current study is cross-sectional and cannot provide evidence on the causal link between domestic work and depression, the cost estimation should be seen as an example on how to estimate the related direct and indirect costs rather than evidence on exact costs that could be avoided if the risk factor of burdensome domestic work would be eliminated in the target population.

## 3. Results

In total, 25% of the women and 14% of the men reported that they spend more than 20 h per week on domestic work (Table 1). The prevalence of spending more than 20 h per week on domestic work was lower in the age group 50–69 years than in the younger age group 30–49 years and among men with a low educational level compared to men with a high educational level. Among women, there was no difference in time spent on domestic work between educational levels. Regarding family status, women who lived with a partner and children and single mothers spent more time on domestic work than others. Non-employed women spent more time on domestic work than employed women.

The proportion who experienced domestic work at least sometimes as burdensome was 57% in women and 39% in men. The unemployed, persons living with a partner and children, single parents and those aged 30–49 years reported more often burdensome domestic work than others (Table 1). Experiencing domestic work as burdensome was most common among unemployed women and both single and non-single mothers. Moreover, 10% of the women and 6% of the men experienced domestic work most or all the time as burdensome (not shown).

The prevalence of self-reported diagnosed depression was 10% in women and 6% in men (Table 1). The prevalence was highest among the unemployed, students, those living alone and single mothers.

There was no overall association between spending more than 20 h per week on domestic work and self-reported diagnosed depression among men, neither before nor after adjusting for age group, educational level, family status, employment status, economic difficulties, and social support (Table 2). Among women, a relatively weak inverse association was found in the fully adjusted model. A strong association was, however, found between experiencing domestic work sometimes or more often as burdensome and depression in both men and women. Adjustments for age, educational level, family status, employment status, economic difficulties and social support attenuated the associations to some extent, but they remained strong and statistically significant. Of the potential confounders, economic difficulties, lack of social support and living alone were associated with depression in both men and women, and in women also unemployment and low and medium educational level (not shown).

When non-employed respondents were analysed separately, a strong adjusted odds ratio was observed for the association between burdensome domestic work and depression (OR = 3.45, 95% CI: 2.65, 4.48 for men and women combined). Moreover, a similar odds ratio was found when parents with children were excluded from the analysis of the total study population (OR = 3.04, 95% CI: 2.55, 3.64 for men and women combined).

Table 3 shows the population attributable risks for experiencing domestic work at least sometimes as burdensome. In total, 44% of the total prevalence of self-reported diagnosed depression could be related to burdensome domestic work in women. The corresponding proportion in men was 51%.

Table 4 presents the total costs of depression and the estimated costs of this disease related to domestic work per year for men and women aged 30–69 years in Mid-Sweden. The total cost of depression per individual per year possibly linked to domestic work was estimated at EUR 2209.1 on average (min EUR 340.0–max EUR 342,777.7). This translates to EUR 135.1 million (min EUR 20.7 million–max EUR 21.4 billion) in the total cost of depression related to domestic work for the whole population in this age range in Mid-Sweden. The corresponding total cost for depression was EUR 286.4 million. Moreover, the cost of depression in general and cost of depression related to domestic work were higher in women (EUR 72.7 million) in comparison to men (EUR 62.3 million).

## 4. Discussion

As expected, the prevalence of spending more than 20 h per week on domestic work was higher in women (25%) than in men (14%) in this study population aged 30–69 years. About six out of ten women and four out of ten men experienced domestic work at least sometimes as burdensome. Experiencing domestic work as burdensome was most prevalent among unemployed women and among single and non-single mothers. A strong independent association was found between experiencing domestic work sometimes or more often as burdensome and self-reported diagnosed depression, both in men and women. According to the estimates, up to almost half of the prevalence, and costs, of depression might be related to burdensome domestic work.

Discrepancies in the association between domestic work and mental health in some previous studies can be due to several factors [1,10,11,14,15,19]. The studies have varied in the stage of the family life course and employment status, e.g., whether only parents were included parents, whether participants had small or more grown-up children, whether single parents were included, whether only employed parents were included and so on. Persons without children have very rarely been included in these studies. Another possibly contributing factor to the discrepancies is the type of domestic work performed. Low-schedule-control tasks, such as laundry and cooking, must be typically performed daily and at certain times, whereas high-schedule-control tasks, such as yard work and car maintenance, can often be performed without any time urgency [1]. Women often spend more hours on low-schedule-control tasks, whereas men spend more hours on high-schedule-control tasks.

In our study, no overall association between the number of hours spent on domestic work and depression was observed in men. This is in line with other studies in Sweden, showing no association between time spent on domestic work and psychological distress [11,17]. A moderate inverse association between time spent on domestic work and depression was obtained in the fully adjusted model in women. This may be due to reverse causality, i.e., that depressed women are not able to spend long hours on domestic work, or that given the same level of experiencing domestic work as burdensome, it is better to have more time to do it. The type of domestic work and in particular the fact that women often spend more hours on low-schedule-control tasks whereas men spend more time on high-schedule-control tasks may contribute to the difference between men and women in the association between the number of hours spent on domestic work and depression. An interesting finding was also that the prevalence of spending more than 20 h a week on domestic work was higher in men with high education than in men with low education, whereas no difference between educational levels was observed in women.

In the current study, a strong independent association was found between experiencing domestic work at least sometimes as burdensome and self-reported diagnosed depression, both in men and women. This finding agrees with several studies showing an association between strain/demands in domestic work and mental health in women [10,13,15,17,19] and in men [10,13,17]. Most of the studies on domestic work and mental health have been cross-sectional, but the study of Melchior et al. found in the prospective French GAZEL study [10] that multiple work and family demands had a strong effect on mental health in both men and women, and particularly on depression. Family demands were measured by the number of dependents in that study. Similarly, Coklin et al. [14] reported in a longitudinal study that work-family conflict was associated with mental health in both men and women with children. Further, they showed that when the work-family conflict was relieved, both mothers’ and fathers’ mental health improved significantly. Both these longitudinal studies were, however, limited to working populations and did not measure time spent on domestic work or strain in domestic work.

Domestic work has rarely been investigated among non-employed persons, but for example Maeda et al. [15] found that domestic work stress was associated with self-rated psychological health in women regardless of employment status. Unemployed women were one of the groups in the present study in which experiencing domestic work as burdensome was most prevalent. The association between burdensome domestic work and depression was also strong when non-employed persons were analysed separately. This indicates that the problem is not limited to those employed and suggests that the theory of work-family conflict is not enough to explain the effects of domestic work on mental health. We were not able to test the different theories concerning domestic work and mental health, but the results are in line with the concept of role strain or overload [1].

In general, domestic work has a large impact on the everyday life of millions of people. Yet, in contrast to paid work, few studies have investigated the association between unpaid domestic work and mental health. The WHO report on social determinants of health [2] recommends that unpaid work, mostly performed by women, should be included in the national accounts as a first step to make it visible. Experiencing domestic work at least sometimes as burdensome was highly prevalent in the study population, and the odds ratios linking it to depression were very high. As a result, the population attributable risks indicated that burdensome domestic work may contribute up to about half of the prevalence of self-reported depression in the population 30–69 years of age. Population attributable risks should, however, be interpreted with caution since several risk factors may occur simultaneously, and the sum of the percentages explained can exceed 100%. Social factors such as economic difficulties, lack of social support and unemployment have been shown to be determinants of poor mental health in previous studies [29] and are therefore likely to contribute to the prevalence of depression in the population. Nevertheless, the results indicate that domestic work should not be omitted when considering factors that affect mental health in the general population.

The economic burden of depression is high, and higher among women than among men. The results suggest that up to almost half of the costs of depression could possibly be related to burdensome domestic work, somewhat higher in women than in men. Since a literature search did not reveal any relevant analysis on the economic burden of depression attributable to domestic work, the current study seems to be the first one to present data on this important issue, both at the population and individual level, and to highlight the differences between women and men. Previous studies have for example estimated the costs of depression attributable to job stress and workplace bullying [30]. On the other hand, the present analysis is based on aggregated administrative data, e.g., health care costs in one region, and assumptions based on the literature, e.g., on indirect costs. This first analysis is an indication of the potential substantial economic burden of domestic work on the mental health of both sexes, but since the cross-sectional design of the study prevents any conclusions about the causal direction of the association, the results should be interpreted with caution, and further research is needed.

The cost estimates were based on population attributable risks. According to Rockhill et al. [31], estimation of the population attributable risk is of most use from a public health perspective, when the factor of interest is clearly causally related to the end point and when there is consensus that the exposure is amenable to intervention. Therefore, future research should include longitudinal studies that could confirm the causal link between domestic work and depression and show whether relieving burdensome domestic work will lead to a decreased risk of depression. This would provide stronger evidence for the costs of depression attributable to domestic work.

### Limitations

As mentioned earlier, a major limitation of this study is the cross-sectional design, which prevents any interpretations about causality. It is possible that part—or even most—of the findings can be explained by reverse causality, i.e., that depression increases the risk of experiencing domestic work as burdensome. This complicates the interpretation of the population attributable risks since these should be based on longitudinal studies [31]. Reverse causality also increases the risk of endogeneity when estimating costs [32], and it is therefore essential to verify the causal link. Combined work and family demands [10,14] and low control at home [33] have been shown to be a risk factor for poor mental health in longitudinal studies, but there is a lack of longitudinal studies on the effect of domestic work in and of itself.

The type of domestic work, number of children or work stress among employed persons, which may influence the associations observed, were not measured in the current study. The response rate was 43% which may affect the representativeness of the respondents and lead to an underestimation of the prevalence of burdensome domestic work and depression. It may also introduce bias in the examined associations, even though this is not necessarily so [34]. Another limitation regards the dichotomisation of the measures of domestic work, which might reduce the specificity of the data. There are, however, no validated measures to examine domestic work due to the lack of research in this field [1,15]. Diagnosed depression was self-reported, and the prevalence was somewhat lower (9%) than the proportion using antidepressants (12%) in this age range in 2017 in Sweden [35]. Yet another limitation is that the estimation of direct and indirect costs was based on administrative data and the literature, not on data on direct costs of the participants in the study.

One of the advantages of the present study is that it is based on a considerable sample of the general population in a large geographical area and represents a wide age group of men and women aged 30–69 years. Although the study was limited to five counties, it covers the general adult population in these counties, comprising about 1 million inhabitants. Another advantage is that the study population includes both men and women, parents and persons without children as well as employed persons and persons outside the labour force. We could also analyse the association between domestic work and depression separately in the subgroups of non-employed persons and persons without children as well as adjust for several socioeconomic and psychosocial factors. Furthermore, we were able to address the relevant economic burden of depression related to domestic work—even though the analyses should be interpreted more as an example on how to estimate the related direct and indirect costs rather than evidence on exact costs.

## 5. Conclusions

In conclusion, experiencing domestic work at least sometimes as burdensome was highly prevalent among women and men in this population-based study. It was also highly associated with self-reported diagnosed depression, and this association was not restricted to the employed or to parents with children. Even though the cross-sectional design does not allow one to assess the direction of the association between domestic work and depression, and longitudinal studies are needed, the results suggest that the prevalence of depression in the general population and its high societal costs are possibly linked to burdensome domestic work. Since domestic work is affecting numerous people worldwide, more effort should be applied to investigate the mental health effects of unpaid domestic work, to improve the theoretical basis and methods to measure different types and aspects of domestic work and to verify the causal link between domestic work and depression. This is necessary in order to be able to design preventive activities with the aim to relieve the burden of depression for individuals and to decrease related societal costs.

## Figures and Tables

**Table 1 ijerph-18-09778-t001:** Number of respondents, proportion spending more than 20 h on domestic work per week, burdensome domestic work, and prevalence of self-reported diagnosed depression in different socio-demographic groups.

Variable	Category	N	N	Spends More than 20 h/week in Domestic Work (%)		Experiences Domestic Work Sometimes or More Often as Burdensome (%)		Self-Reported Diagnosed Depression (%)	
		Women	Men	Women	Men	Women	Men	Women	Men
Total		7981	6203	25.4	14.0	56.8	38.8	10.5	6.5
Age group	30–49 years	3241	2321	33.8	21.6	68.8	51.4	12.8	7.1
	50–69 years	4740	3882	19.7	9.4	48.6	31.3	8.9	6.2
Educational level	Low	675	820	24.1	9.8	51.0	31.7	15.2	7.5
	Medium	3567	3030	25.9	13.4	55.0	38.7	11.7	7.0
	High	3712	2326	25.2	16.2	59.8	41.4	8.5	5.5
Family status	Living alone	1121	959	11.5	6.4	47.1	40.4	16.0	12.2
	Living with partner	3507	2734	19.9	9.5	46.2	26.8	8.5	4.8
	Living with partner and children	2673	2068	37.7	23.5	72.1	52.6	9.3	4.3
	Single parent	528	270	32.2	17.3	70.1	52.6	16.9	11.0
Employment status	Employed	6182	4964	23.5	13.8	57.7	39.5	8.8	4.9
	Student	179	119	32.4	24.4	62.0	49.6	18.5	12.8
	Unemployed	156	131	30.6	10.4	72.4	45.0	27.6	16.2
	Other	1464	989	32.2	13.9	51.1	33.4	15.1	12.8

**Table 2 ijerph-18-09778-t002:** Univariate and multivariate odds ratios (95% confidence interval in parenthesis) for self-reported diagnosed depression among women and men aged 30–69 years.

	*Women*		*Men*	
	Unadjusted ^1^	Adjusted ^2^	Unadjusted ^1^	Adjusted ^2^
Time spent in domestic work				
0–20 h/week	1 (reference)	1 (reference)	1 (reference)	1 (reference)
21+ h/week	0.95 (0.81, 1.13)	0.77 (0.64, 0.93)	1.00 (0.75, 1.35)	0.99 (0.71, 1.37)
Experiences domestic work as burdensome				
Never/seldom	1 (reference)	1 (reference)	1 (reference)	1 (reference)
Sometimes or more often	2.80 (2.36, 3.32)	2.40 (1.99, 2.88)	3.97 (3.17, 4.97)	3.68 (2.88, 4.72)

^1^ Crude odds ratio. ^2^ Time spent on domestic work and burdensome domestic work in the same model and adjusted for age group, educational level, family status, employment status, economic difficulties, and social support.

**Table 3 ijerph-18-09778-t003:** Prevalence (%) of burdensome domestic work, odds ratio (OR) and population attributable risk (PAR) for self-reported diagnosed depression among women and men aged 30–69 years.

	Experiences Domestic Work Sometimes or More Often as Burdensome (%)	OR for Self-Reported Diagnosed Depression	PAR
**Women**	56.8	2.40	44.3
**Men**	38.8	3.68	51.0

**Table 4 ijerph-18-09778-t004:** Cost of depression and cost related to domestic work per year for men and women, and in total, aged 30–69 years in Mid-Sweden.

		*Women*			*Men*	
**Cost categories**	**Average (EUR)**	**min (EUR)**	**max (EUR)**	**Average (EUR)**	**min (EUR)**	**max (EUR)**
Health care cost per patient *	1370,1	187,8	309923,2	1662,1	275,6	186915,5
Cost of antidepressants per patient **	139,4	25,2	454,3	139,4	25,2	454,3
Total direct costs per patient	1509,5	213,0	310377,5	1801,4	300,7	187369,8
Indirect costs per patient	2803,3	395,5	576415,3	3345,5	558,5	347972,5
**Cost of depression—per patient per sex per year**	**4312,8**	**608,5**	**886792,8**	**5147,0**	**859,2**	**535342,3**
Total direct costs 30-69 yrs in Mid-Sweden	57476232,6	8109964,3	11818272411,0	42770509,1	7139600,8	4448609214,4
Indirect costs 30-69 yrs in Mid-Sweden	**106741574,8**	**15061362,3**	**21948220191,9**	**79430945,4**	**13259258,7**	**8261702826,7**
**Cost of depression—all population per sex per year**	**164217807,4**	**23171326,6**	**33766492603,0**	**122201454,4**	**20398859,6**	**12710312041,1**
% of population with self-reported depression	**10,5**			**6,4**		
Estimated population with depression	**38077,1**			**23742,4**		
% of depression related to domestic work	44,3			51,0		
Estimated population with risk for depression related to domestic work	**16868,2**			**12108,6**		
**Cost of depression related to domestic work—all population per sex per year**	**72748488,7**	**10264897,7**	**14958556223,1**	**62322741,8**	**10403418,4**	**6482259140,9**
**Total cost of depression related to domestic work—per patient per sex per year**	**1910,6**	**269,6**	**392849,2**	**2625,0**	**438,2**	**273024,6**
		**Total**				
**Cost of depression—all population (men and women combined)**	286419261,8	43570186,2	46476804644,1			
**Total cost of depression per year related to domestic work—all population**	**135071230,4**	**20668316,1**	**21440815364,1**			
**Average cost of depression related to domestic work—per patient per year**	2209,1	340,0	342777,7			

* Aggregated cost data retrieved from Cost-Per-Patient database, Region Värmland. ** Aggregated cost data retrieved from The Dental and Pharmaceutical Benefits Agency (TVL).

## Data Availability

The dataset analysed during the current study is not publicly available due to confidentiality and regulations under the Swedish law (the Public and Privacy Act 2009:400, Chapter 24, Section 8).

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
