# Peer review of "Domestic Work, Self-Reported Diagnosed Depression and Related Costs among Women and Men—Results from a Population-Based Study in Sweden"

_ijerph, 2021, doi:10.3390/ijerph18189778_

Round 1
Reviewer 1 Report
The study focuses on an interesting topic. Even though unpaid domestic work has a large impact on the daily life for many women and men, the effects on health have been much less studied compared to the health effects of paid work. It is also very interesting that the study has analyzed the economic burden of mental health (i.e. depression) due to work tasks related to household and family. The result showing that burdensome domestic work accounted for up to about half of the prevalence of depression among individuals 30-69 years of age was surprising. However, this is a cross-sectional study and several risk factors included in the study may occur simultaneously which makes the results concerning the attributable risk somewhat uncertain. This uncertainty could have been described more clearly also in the conclusion in the abstract.
Author Response
Thank you for your comment and suggestion. We have now elaborated the discussion on the limitations due to the cross-sectional design and underlined this uncertainty also in the conclusion in the abstract.
Reviewer 2 Report
Overall, this is an insightful study that examines the relationship among domestic work, depression, and cost among women and men in Sweden. Here are a few questions/comments for you to consider.
- Given your findings, would different explanations be generated if you also consider the perception of domestic work? For example, perceiving domestic work as temporary/short term vs. long-term; enjoy domestic work as a hobby vs. as responsibility/unpaid jobs? I wonder if this could explain why “no overall association between the number of hours spent in domestic work and depression was observed in men”.
- There is a gap between the cost estimates and actual costs by participants in the study. Could you explain in your limitations?
Wish you the best!
Author Response
Thank you for your encouraging comment and suggestions.
- We agree and have added a comment to the discussion that the difference between men and women in the association between time in domestic work and depression can be due to the type of domestic work and control over tasks.
- We have added a comment on this difference to the limitations.
Reviewer 3 Report
This paper addresses an important question that remains underexplored and uses an original dataset to investigate it. The main theories linking domestic work and depression are mentioned in the paper, although they could be discussed more in-depth in light of the results.
There is a problematic issue that needs to be taken into account more thoroughly: reverse causation. The authors acknowledge this problem, but very briefly and at the end of the paper, and they seem to discard it because other studies point at high loads of domestic work influencing depression. Some studies have pointed at this direction of causality, but the link is between the amount of domestic work and depression, not between the perception of domestic work as burdensome and depression, and therefore endogeneity can be discarded here. In addition, it seems quite plausible that depressed individuals will find domestic chores (or any chore) more burdensome than non-depressed individuals, irrespective of the amount of work they perform. This is important because, although the authors talk about "associations" between the two variables, they also try to estimate the costs of depression attributable to domestic work, and this calculation requires assuming that causality goes one way. This is a very strong and problematic assumption that would need to be discussed. In my opinion, the discussion about costs of depression should be dropped because calculating them in this way assumes that the model can perfectly predict depression, which it can't.
Two additional comments I would like to add:
- I think readers would like to see the whole regression tables and not only the unadjusted and adjusted coefficients for the main variable of interest.
- Some results seem to contradict existing theories, for instance, the fact that unemployed women have higher depression rates/find domestic chores more burdensome. The authors could comment further on the results from all controls in light of the literature. There is a theoretical contribution to be made in this paper in my opinion.
Author Response
Thank you for your comments and suggestions.
We have added some comments on the presented theories to the discussion. We were not able to test the different theories directly, but in general the results are in line with the concept of role strain or overload. We have also pointed out that the work-family conflict theory is not enough to cover the association between domestic work and mental health since the association is not limited to those employed.
We fully acknowledge the limitation of the cross-sectional design of the study and the possibility of reverse causality. We have now highlighted these limitations more clearly in the discussion and revised the conclusions of the study, including the abstract. The referenced longitudinal studies investigated the effects of multiple work and family demands or work-family conflict and are not directly comparable with the present study. Therefore, we have modified the discussion and emphasized that more longitudinal studies are needed in this field in general and specifically to verify the causal link between domestic work and mental health in order to achieve methodologically stronger estimates of the economic burden that can be attributed to domestic work.
Additional comments:
- We agree that the results for additional variables would be interesting to show but then we would be presumed to comment these results in the results and discussion sections, which would unnecessarily extend the content of the paper. Therefore, we decided to focus on the main findings and have only added a comment on the statistically significant results for the confounding variables.
- We have commented these theories (see above) in the discussion. There are several interesting findings for the controls (e.g. for educational level, unemployment), but we feel that these fall outside the main aim of the study and should be studied in more detail in a separate study.
Reviewer 4 Report
The paper "Domestic work, self-reported diagnosed depression and related costs among women and men - Results from a population-based study in Sweden analyzes a very important issue in all countries. However, there are some aspects that need to be improved.
1. In the abstract, the authors should briefly comment on the political implications of these results.
2. In the Introduction, the contribution of the article in the literature should be improved. Specifically, why is this peper different from the other studios?
3. Materials and Methods is poor. There is no justification for the 30-69-year-old sample. Not specified and not described Multivariate logistic regression models.
4. Improve the presentation and interpretation of results.
5. The conclusions must be improved. What do these findings imply? How can you help the population?
Author Response
Thank you for your comments and suggestions.
- We have revised the conclusions in the abstract and stated that it is important to take domestic work into account when considering factors that contribute to the prevalence of depression in the general population and the high societal costs of depression.
- We have added to the introduction why it is important to study the association between domestic work and mental health in the general population (not only among employed people or among individuals with children) and related costs. The contribution of the present study is also highlighted in the discussion.
- We have revised the text in the material and methods section to better describe why only the age group 30-69 years was available for this study and provided a more exact description of the multivariate logistic regression models.
- We have made some changes in the text in the results section and in the tables to clarify the results.
- We have revised the conclusions as this was brought up by all reviewers. In general, more longitudinal studies are required and there is a need to improve the theoretical basis and methods to measure different types and aspects of domestic work. This is important to be able to design preventive activities with the aim to relieve the burden of depression for individuals and to decrease related societal costs.
Round 2
Reviewer 3 Report
This new version of the paper discusses existing theories more and uses the results to contribute to them more clearly. However, I think the endogeneity issue is still there, and simply acknowledging it is not enough.
As I mentioned in my first review, you need to assume one direction of causality in order to calculate the costs, and attributing depression costs to the perception of burdensome work does not seem justified. Otherwise, we could also attribute the costs of depression to any variable with a positive and significant coefficient in the model, like unemployment. Perceiving domestic work as burdensome can be the result of depression, rather than one of its causes. Unless the authors provide empirical evidence that points towards one causal direction, both remain plausible (and the calculation of the costs becomes impossible).
I also think the authors need to include the full regression tables, even if they only provide detailed comments on the variable of interest, for transparency.
Author Response
Thank you for your further comment.
As you indicated we acknowledge the issue of endogeneity due to the possibility of reverse causality and have now stated it explicitly in the limitations with a new reference (Aarstad). We have also pointed out the limitations due to the cross-sectional study design, the need for longitudinal studies in this research field, and that one should interpret the estimation of costs more of an example on how to calculate costs rather than an estimation of actual costs in the manuscript. However, we realize that use of the term “attributable” when discussing costs was misleading and we have thus reformulated all the sentences referring to the costs of depression related to domestic work in this study (see also our answer to Academic Editor). We have made an additional literature search for longitudinal studies in this field but could only find one study by Griffin et al. which shows that low control at home is a risk factor for depression. All the three longitudinal studies (Melchior et al, Coklin et al and Griffin et al) suggest that there is a causal link between domestic work and mental health, but the evidence is only indirect since these studies have used different measures and there is a lack of standardized methods to measure domestic work (as pointed out by us and other researchers). We have also consulted an additional health economist for advice and added a comment to the limitations that the population attributable risks should also be based on longitudinal studies and made some further adjustments to the conclusions.
Among the covariates, economic difficulties (OR=2.96, 95% CI: 2.24, 3.92 in men and OR=3.25, 95% CI: 2.69, 3.93 in women) and lack of social support (OR=2.34, 95% CI: 1.81, 3.04 in men and OR=2.20, 95% CI: 1.77, 2.75 in women) were the most strongly associated factors with depression. Due the extensive length of the manuscript and the fact that these factors are already known determinants of mental health and therefore not the focus of the study, we have adjusted for them in the analyses but not included them in the tables. We are however willing to add the full regression tables if the editor finds important that these should be provided. We have also added a new reference (Alegría et al.) where more information about other social determinants of mental health can be found.
Reviewer 4 Report
The authors have improved the paper, and I think it could be accepted.
Author Response
Thank you for your encouraging comment.